# Potential of New Bacterial Strains for a Multiproduct Bioprocess Application: A Case Study Using Isolates of Lactic Acid Bacteria from Pineapple Silage of Costa Rican Agro-Industrial Residues

**Jéssica Montero-Zamora [1], María Daniela Rojas-Vargas [2,3], Natalia Barboza [4,5], José Pablo López-Gómez [1,6], José Aníbal Mora-Villalobos [1] and Mauricio Redondo-Solano [2,3,*]**

[1] National Center for Biotechnological Innovations of Costa Rica (CENIBiot), San Jose 1174-1200, Costa Rica; jessica.monterozamora@gmail.com (J.M.-Z.); plopezgomez@atb-potsdam.de (J.P.L.-G.); anibalmora@gmail.com (J.A.M.-V.)

[2] Tropical Disease Investigation Center (CIET), Department of Microbiology and Immunology, Faculty of Microbiology, University of Costa Rica, San Jose 11501-2060, Costa Rica; danirova.dr@gmail.com

[3] Food Microbiology Research and Training Laboratory (LIMA), Department of Microbiology and Immunology, Faculty of Microbiology, University of Costa Rica, San Jose 11501-2060, Costa Rica

[4] Food Technology Department, University of Costa Rica, San Jose 11501-2060, Costa Rica; natalia.barboza@ucr.ac.cr

[5] National Center for Food Science and Technology (CITA), University of Costa Rica (UCR), San Jose 11501-2060, Costa Rica

[6] Microbiome Biotechnology Department, Leibniz Institute for Agricultural Engineering and Bioeconomy (ATB), 14469 Potsdam, Germany

* Correspondence: mauricio.redondosolano@ucr.ac.cr

**Abstract:** Lactic acid bacteria (LAB) with potential for the development of multi-product processes are necessary for the valorization of side streams obtained during the biotechnological production of lactic acid (LA). In this study, 14 LAB strains isolated from pineapple agro-industrial residues in Costa Rica were cultivated in microplates, and the six strains with the highest growth were selected for fermentation in microbioreactors to evaluate the production of LA and acetic acid, and the consumption of glucose. *Lacticaseibacillus paracasei* 6710 and *L. paracasei* 6714 presented the highest $OD_{600}$ values (1.600 and 1.602, respectively); however, the highest LA (in g/L) production was observed in *L. paracasei* 6714 (14.50 ± 0.20) and 6712 (14.67 ± 0.42). *L. paracasei* 6714 was selected for bioreactor fermentation and reached a maximum $OD_{600}$ of 6.3062 ± 0.141, with a LA yield of 84.9% and a productivity of 1.06 g $L^{-1}$ $h^{-1}$ after 21 h of fermentation. Finally, lipoteichoic acid (LTA) detection from biomass was performed and the antimicrobial activity of the compounds present in the supernatant was studied. LTA was detected from *L. paracasei* 6714 biomass, and its supernatant caused significant inhibition of foodborne surrogate microorganisms. LAB isolated from pineapple silage have biotechnological potential for multiproduct processes.

**Keywords:** lactic acid; lipoteichoic acid; antimicrobial compounds; fermentation; circular economy; bioproducts valorization

## 1. Introduction

Lactic acid bacteria (LAB) are microorganisms defined as non-sporulating, acid-tolerant, catalase-negative, Gram-positive cocci or rods; they are characterized by the production of lactic acid (LA) as the major end-metabolic-product of carbohydrate fermentation [1–3]. LA production from LAB is a growing business given the broad applications of this compound, ranging from the food to the pharmaceutical industries, with a currently growing market based on the production of polylactic acid [4]. The worldwide demand for LA is expected to grow to a market value of $9.8 billion by 2025 [5]. However, the

downstream processing (DSP) of LA accounts for 40–70% of the total production cost; thus, there is a need to develop a convenient and economic strategy for the exploitation of secondary products, such as antimicrobial peptides and cell biomass [6]. Among the products of biotechnological interest that could be simultaneously obtained with LA are other organic acids, antimicrobial substances (bacteriocins), and compounds derived from the bacterial cell wall [7].

Lactic acid bacteria produce bioactive peptides or proteins that display bactericidal activity, mainly against Gram-positive bacteria, which could be useful for food applications, among others [8]. Given that LA has poor solubility in organic solvents, a convenient separation strategy (based on organic extraction) could be applied to easily separate bacteriocins from this compound [7]. This is relevant as previous studies have shown that the overall recovery of LA yield from complex fermentation media is about 45%, which means that there is a large volume of fermentation broth considered as waste or losses [9]. The recovery of bacteriocins may be a suitable alternative for adding value to other metabolites in the fermentation media.

A similar situation is present in the case of cell biomass, which is normally treated as a by-product in most biotechnological processes (except for the industrial production of probiotics) [7]. LAB cells contain different compounds with potential commercial value, such as lipoteichoic acid (LTA), which is contained in the cell wall of Gram-positive bacteria [10,11]. At the beginning of the DSP (where cells are discarded), LTA could be separated, while the supernatant follows the pipeline. LTA molecules are composed of a covalently linked glycolipid and hydrophilic polymer of glycerophosphate [12,13]. Different studies have reported LTA structural and functional variations according to the genera and species of bacteria [14]. LTA has a wide range of potential applications, such as the development of new food, pharmaceutical, and cosmetic products [15].

As LAB have the ability to metabolize different substrates for the simultaneous production of highly valuable compounds different to LA, they can be considered as good candidates for multi-product bioprocess applications and the development of integrated biorefineries [16]; these are defined as multi-product processes that are based on the conversion of renewable materials into bio-based products. Integrated biorefineries are considered as a practical way to improve industrial single-product biotechnological processes as they allow the valorization of residues [17]. The isolation, study, development, and application of robust microbial strains are pivotal in the development of biorefineries [18]. LAB strains with biotechnological potential are increasingly isolated from non-traditional biorefinery feedstocks [19], such as non-dairy fermented foods, vegetables, fruit juices, and vegetable waste [8,20]. As the pineapple industry in Costa Rica generates an important amount of organic residue, it is possible that it can also serve as an interesting source of new strains with biotechnological potential.

Given the above, this research explores the general metabolic profile of LAB strains isolated from pineapple agro-industrial residues in Costa Rica to establish their potential use for the development of a multi-product process for LA, antimicrobial compounds, and LTA.

## 2. Materials and Methods

### 2.1. Bacterial Strains and Growth Conditions

2.1.1. Lactic Acid Bacteria

Fourteen LAB isolated from silage pineapple peel residues in Costa Rica were selected for this research [21]. These isolates belong to the collection of the National Center for Food Science and Technology (CITA) at the University of Costa Rica, and they include both homofermentative and heterofermentative microorganisms; all isolates were considered for the experiments, as this is a preliminary study aimed to identify bacteria with potential for integrated biorefineries. The isolates were identified as follows: *L. paracasei* 6709, *L. paracasei* 6710, *L. paracasei* 6711, *L. paracasei* 6712, *L. paracasei* 6713, *L. paracasei* 6714, *L. paracasei* 6715, *Limosilactobacillus fermentum* 6702, *L. fermentum* 6704, *Lentilactobacillus parafarraginis* 6717,

*L. parafarraginis* 6719, *Weissella ghanensis* 6706, *Fructobacillus tropaeola* 6705, and *F. tropaeoli* 6707. All the bacteria were kept with 20% (*v/v*) glycerol at $-80 \pm 2$ °C in De Man, Rogosa, and Sharpe broth (MRS) (Difco, Le Pont de Claix, France). Precultures of each LAB were prepared in MRS broth and incubated overnight at $37 \pm 1$ °C for 24 h under static conditions.

### 2.1.2. LAB Growth Curves

Growth curves for each of the isolates were determined to define the growth kinetic parameters (µ max). The bacterial suspensions of each strain were diluted in fresh MRS broth to attain an absorbance of 0.05 at 600 nm ($OD_{600}$). A 96-well microplate (Fisher Scientific, Bridgewater, NJ, USA) was filled with a constant volume (250 µL) of each bacterial suspension in quintuplicate. The growth kinetics were assessed by using the same incubation conditions reported above; bacterial growth was determined by measuring the $OD_{600}$ every 15 min using a microplate reader (Sinergy HT, Biotek, Winooski, WI, USA).

### 2.2. Production of Metabolites

For the determination of metabolites of interest, i.e., LA, acetic acid (AA), and antimicrobial compounds, the strains showing the best growth performance were grown in MRS broth with an initial absorbance ($OD_{600}$) of 0.05 and initial pH value of 6.8; the broth was incubated at $37 \pm 1$ °C for 24 h under static conditions. The fermentation was conducted in a microbioreactor (Applikon Biotechnology, Delft, The Netherlands) with a fermentation volume of 6 mL in triplicate. After fermentation, samples of each tube were centrifuged at $10,000 \times g$ for 5 min at room temperature. The supernatant was recovered and filtered using a 0.20 µm-pore-size syringe filter composed of regenerated cellulose; it was used for the quantification of organic acids and glucose by high-performance liquid chromatography (HPLC; system Shimadzu, Columbia, MD, USA) equipped with an autosampler (Shimadzu, SIL-20A HT, Columbia, MD, USA), as well as for antagonistic activity assays.

### 2.3. Bioreactor Fermentation

One selected strain was fermented in a 7 L stirred-tank batch (Applikon, Delft, The Netherlands) with an operating volume of 4 L, 0.5 vvm, 150 rpm, pH of 6.8 (maintained by the automatic addition of 20 % NaOH), and temperature of $37 \pm 1$ °C for 30 h. Fermentation was evaluated by periodic sampling every 2 h (10 mL). For microbial quantification, the $OD_{600}$ of the samples was measured using a spectrometer (Lambda 35, Perkin Elmer, Waltham, MA, USA). Additionally, the samples were centrifuged at $10,000 \times g$ for 5 min at room temperature and the supernatant was filtered using a 0.20 µm-pore-size syringe filter composed of regenerated cellulose and frozen at $-20$ °C. Lactic acid production and residual sugars were analyzed by the HPLC system (Shimadzu, Columbia, MD, USA). LTA was determined at the end of the fermentation. The LA yield and maximum productivity were estimated in terms of grams of lactic acid per liter of fermentation medium per hour [22].

### 2.4. Chemical Analytical Methods

### 2.4.1. HPLC Analysis

For the quantification of organic acid, 1 mL of sample was used. The detection was performed using HPLC and a photodiode array detector (Shimadzu, SPD-M20AV, 210 nm). The cleaning of the sample was conducted using Oasis HLB cartridges (Waters, Milford, MA, USA) and it was then filtered through a 0.45 µm membrane filter prior to an injection using a Hi-Plex H column (Agilent, 8 µm, $300 \times 7.8$ mm). The column temperature was 60 °C and the mobile phase (2.25 mM sulfuric acid) had a flow rate of 0.5 mL/min; the injection volume was 5 µL [23].

A HPLC system (1260 infinity, Agilent, Santa Clara, CA, USA), equipped with an autosampler, and a refractive index detector (Agilent, G1362 A, Santa Clara, CA, USA) were used to analyze residual sugars. The sample was extracted using the method described in ISO 11868:2007 [24]; the final mixture was filtered through a 0.45 µm membrane filter and analyzed using a Zorbax Carbohydrate column (Agilent, 5 µm, $4.6 \times 150$ mm). The

following experimental settings were used for the analysis: column temperature of 30 °C, flow rate of the mobile phase (75:25 acetonitrile:water) of 1.2 mL/min, and injection volume of 5 μL. For the quantification of the analytes, a standard curve was built and the LabSolutions and OpenLab CDS software was used for the integration of the peaks.

### 2.4.2. LTA Analysis

Lipoteichoic acid was purified after fermentation according to the procedure described by Morath et al. [25] with some modifications. Briefly, the biomass ($3000 \times g$) from each bioreactor was concentrated by centrifugation at $2000 \times g$ for 10 min and washed three times with phosphate-buffered saline (PBS). The biomass pellet was mixed with an equal volume of n-butanol at 400 rpm for 10 min at room temperature. After centrifugation, the aqueous phase was lyophilized (Christ, Osterode am Harz, Germany) [26,27].

### 2.4.3. Polyacrylamide Gel Electrophoresis (PAGE) Analysis of LTA

Resolving gels (0.75 mm thickness) containing 20% (wt/vol) of 30/6% acrylamide/bisacrylamide were prepared in a 0.25 M Tris-borate buffer (pH 8.2). A total of 25 mg of the LTA extract was mixed with 1 mL of deionized water, and 20 μL of this mixture was loaded onto the gel. The LTA samples were analyzed using a Bio-Rad Protean II Xi electrophoresis cell. The gels were run for 2 h at 80 mA/gel with a buffer containing 0.1 M tris base and 0.1 M tricine at pH 8.2. The gels were fixed and stained with 5 ppm Alcian blue in ethanol and AA (40% and 5% in water) at 25 °C overnight. Until this point, LTA bands were not visible; to show the staining pattern, the Bio-Rad silver stain kit (161-0449 Bio-Rad, Hercules, CA, USA) was used according to the manufacturer's instructions [28,29]. *Staphylococcus aureus* was used as a positive LTA control.

### *2.5. Microbiological Analysis*

#### 2.5.1. Surrogate Bacteria and Growth Conditions

Before starting the evaluation of antimicrobial compounds, each bacterium and surrogate strain was grown in MRS or Tryptic Soy Broth (TSB) (Oxoid, Basingstoke, UK) at $35.0 \pm 0.5$ °C for $24 \pm 2$ h, respectively.

#### 2.5.2. Antimicrobial Activity Determination

The antimicrobial activity of the supernatant of selected LAB strains against *Escherichia coli* ATCC 25922, *Listeria innocua* 4.1, *L. innocua* 5.1, and *Pseudomonas fluorescens* 6.2, isolated from meat products, was determined as described previously [30]. Briefly, each sample was centrifuged at $10,000 \times g$ for 15 min. The supernatant was filtered (0.2 μm) using sterile test tubes and the pH of each sample was adjusted to $7.0 \pm 0.2$ using 2 M NaOH to avoid an inhibitory effect due to LA exposure. Individual tests were performed in each well of a 96-well microplate by mixing 50 μL of sterile TSB, 50 μL of each surrogate solution, and variable ratios (1:1, 1:2, 1:4, and 1:8) of filtered supernatant adjusted with sterile MRS. The positive controls consisted of the mixture without the filtered supernatant (included sterile media instead), while negative controls included just fresh media (instead of bacterial suspension). The incubation of the microplates was conducted inside a humid chamber at $37.0 \pm 0.5$ °C for 24 h and $OD_{600}$ was measured in a microplate reader. All determinations were performed in triplicate.

### *2.6. Data Analysis and Statistical Design*

#### 2.6.1. Bacterial Growth

Statistical analysis was carried out by an analysis of variance and Dunnett's post hoc test to determine significant differences ($p < 0.05$) with the JMP Software package V.15 (SAS Institute Inc. 2019, Cary, NC, USA).

### 2.6.2. Antimicrobial Activity

A two-way analysis of variance (ANOVA) followed by Tukey's honestly significant difference test were used to determine the antimicrobial effect of the supernatant solutions on the test bacteria ($p$-value of <0.05). The analysis was performed using the JMP Software package V.15 (SAS Institute Inc. 2019).

## 3. Results and Discussion

### 3.1. Selection of LAB Strain for Bioreactor Fermentation

The optical density values of the 14 isolates after 24 h of incubation ranged from 0.470 to 1.602. The isolates *L. paracasei* 6710 and *L. paracasei* 6714 presented the highest $OD_{600}$ values (1.600 and 1.602 respectively), followed by *F. tropaeoli* 6707, *L. paracasei* 6709, *W. ghanensis* 6706, and *L. fermentum* 6704, with $OD_{600}$ of 1.081, 1.099, 1.107, and 1.196, respectively. The differences in the $OD_{600}$ values between the isolates with the highest growth and the other strains were statistically significant ($p < 0.05$) (Figure 1 and Table 1).

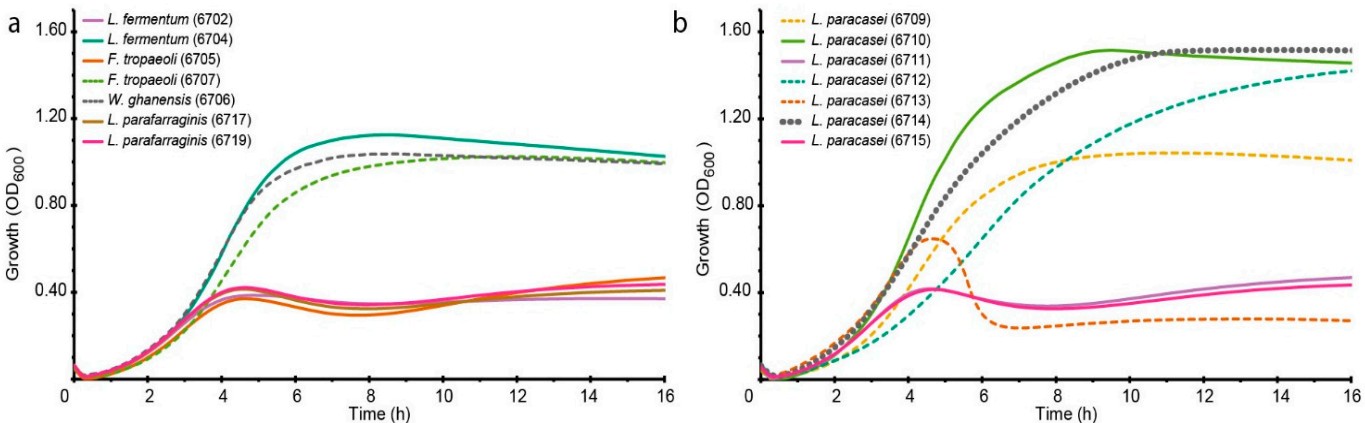

**Figure 1.** Growth curves of lactic acid bacteria (LAB) isolates obtained from pineapple silage residue in De Man, Rogosa, and Sharpe broth (MRS) media. (**a**) LAB strains different to *L. paracasei*; (**b**) LAB belonging to the *L. paracasei* group.

**Table 1.** Maximal optical density ($OD_{600}$) and specific growth rates of lactic acid bacteria (LAB) strain growth kinetics in De Man, Rogosa, and Sharpe broth (MRS) media.

| Strain | Maximal OD$_{600}$ * | Growth Rate (h$^{-1}$) * |
|---|---|---|
| *F. tropaeoli* 6705 | 0.54 7 ± 0.149 [a] | 0.129 ± 0.049 [a] |
| *F. tropaeoli* 6707 | 1.081 ± 0.239 [b] | 0.250 ± 0.100 [b,c,d,e] |
| *L. paracasei* 6709 | 1.099 ± 0.224 [b] | 0.250 ± 0.104 [c,d,e] |
| *L. paracasei* 6710 | 1.600 ± 0.215 [c] | 0.366 ± 0.046 [e] |
| *L. paracasei* 6711 | 0.555 ± 0.052 [a] | 0.145 ± 0.043 [a,b] |
| *L. paracasei* 6712 | 1.015 ± 0.165 [b] | 0.191 ± 0.029 [a,b,c,d] |
| *L. paracasei* 6713 | 0.703 ± 0.152 [a] | 0.242 ± 0.051 [a,b,c,d,e] |
| *L. paracasei* 6714 | 1.602 ± 0.248 [c] | 0.266 ± 0.115 [d,e] |
| *L. paracasei* 6715 | 0.504 ± 0.047 [a] | 0.141 ± 0.040 [a] |
| *L. fermentum* 6702 | 0.484 ± 0.066 [a] | 0.137 ± 0.052 [a] |
| *L. fermentum* 6704 | 1.196 ± 0.226 [b] | 0.306 ± 0.060 [d,e] |
| *L. parafarraginis* 6717 | 0.470 ± 0.043 [a] | 0.147 ± 0.044 [a,b,c] |
| *L. parafarraginis* 6719 | 0.501 ± 0.034 [a] | 0.147 ± 0.045 [a,b,c] |
| *W. ghanensis* 6706 | 1.107 ± 0.164 [b] | 0.283 ± 0.009 [d,e] |

* Data are expressed as the mean ± standard deviation of values obtained from triplicates of the experiments. Different letters in a column mean significant differences.

The six strains with the highest growth were selected for fermentation in micro-bioreactors to evaluate the production of LA and AA, and glucose consumption. The

highest amounts of LA (in g/L) were observed in *L. paracasei* strains 6714 (14.50 ± 0.20), 6712 (14.67 ± 0.42), and 6713 (15.10 ± 0.30), whereas the highest AA producers (in g/L) were *L. paracasei* 6709 (4.62 ± 0.32), *W. ghanensis* 6706 (5.04 ± 0.54), and *F. tropaeoli* 6707 (5.27 ± 0.87) (Figure 2). These results were not surprising given that *L. paracasei*'s capacity to produce high amounts of LA has been reported before [31]. In the case of *F. tropaeoli*, a recent study by Semsik et al. [32] established that this species is a good AA producer, and, in some cases, the values of this metabolite are even higher than those of LA. The observed differences can be explained in terms of the fermentation pathway followed by the different species, i.e., homofermentative and heterofermentative; glycolysis and the primary production of LA are based on the former. On the other hand, the second pathway, also called the pentose phosphate pathway, is characterized by the production of $CO_2$, ethanol, or acetate in addition to LA [23]; this is characteristic of *W. ghanensis* and *F. tropaeoli*. *L. paracasei* species have been reported as a facultative heterofermentative bacteria [33]; therefore, they can follow any of the routes, depending on the growth conditions. All the strains from the study were able to produce AA, meaning that they followed the heterofermentative route, although with a variable LA/AA ratio; *L. paracasei* strains favored the production of LA in all cases.

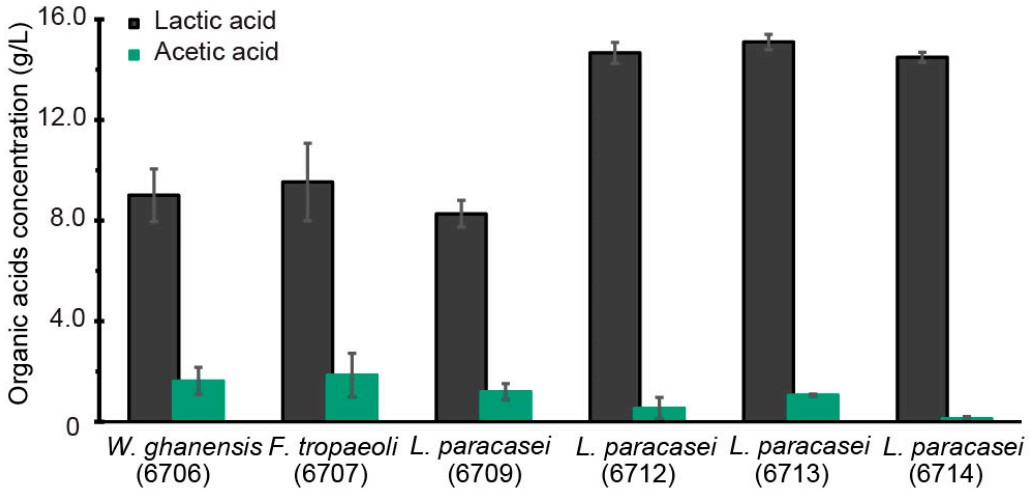

**Figure 2.** Organic acid production by six selected LAB strains during microbioreactor fermentation with an operating volume of 6 mL, initial pH of 6.8, and 37 °C. The values are shown as the mean ± standard deviation, n = 3. The specific LA enantiomer was not determined.

The strain *L. paracasei* 6714 was selected for bioreactor fermentation due to its capacity to combine higher LA production with a higher growth rate. The bioreactor fermentation had a total working volume of 4 L per bioreactor. The batch had an initial LA concentration of 7.8 g $L^{-1}$ and a glucose concentration of 17.5 g $L^{-1}$. The final volume of the fermentation was 4.2 L (considering NaOH addition and sampling), with a LA concentration of 15.7 g $L^{-1}$ (without blank). After 14 h of fermentation, the culture in the bioreactor reached an OD600 of 6.3062 ± 0.141, with a LA yield of 84.9% and a productivity of 1.06 g $L^{-1}$ $h^{-1}$. At the end of the fermentation process (21 h), the bioprocess reached a LA yield of 89.6%, with a productivity of 0.76 g $L^{-1}$ $h^{-1}$ and a maximum OD600 of 6.1627 ± 0.143 (Figure 3). In a recent study, Han et al. [34] determined the capacity of nine *L. paracasei* strains to produce LA in MRS medium. The authors observed that LA production ranged from 15.3 to 17.7 g $L^{-1}$, and the yield was between 76.4 and 88.3%, which is consistent with the observations of this study. The same authors observed a 193% improvement in the LA production capacity of the same strains after optimizing the culture conditions. Similarly, *L. paracasei* subsp. *paracasei* CHB2121 was able to produce 192 g/L lactic acid from medium containing 200 g/L of glucose, with 3.99 g/(L·h) productivity and 0.96 g/g yield; the composition in this case was different to that of standard MRS medium [35]. These results

confirm that the strain *L. paracasei* 6714 fulfills the basic metabolic profile for LA production, and higher performance may be obtained after the optimization of culturing conditions.

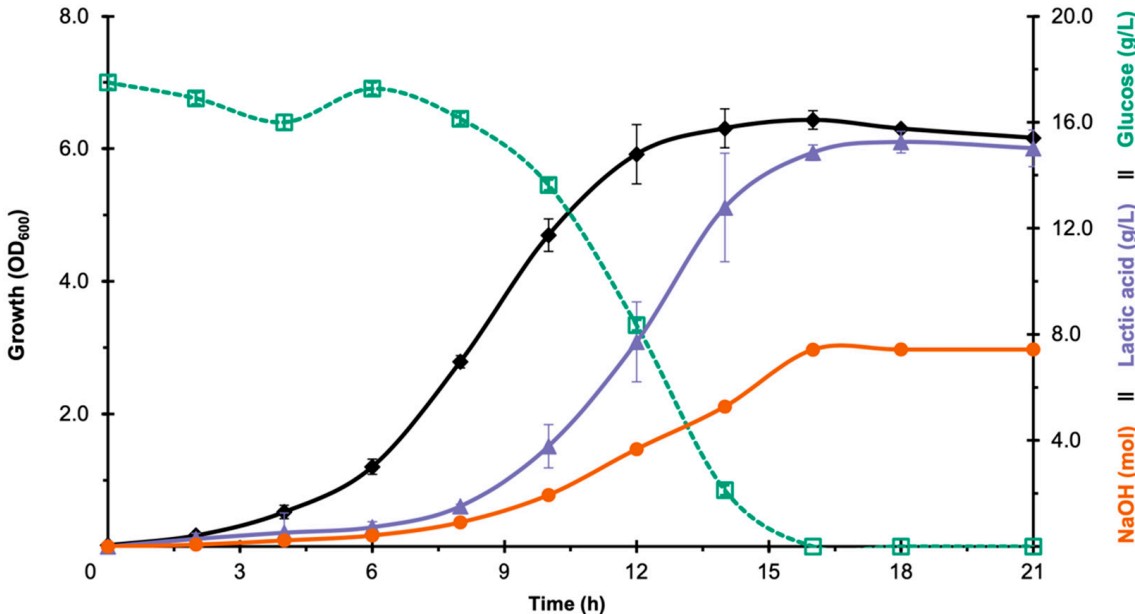

**Figure 3.** Kinetics of strain *Lacticaseibacillus paracasei* 6714 in De Man, Rogosa, and Sharpe broth (MRS) media in a 7 L stirred-tank batch bioreactor with an operating volume of 4 L, 0.5 vvm, pH of 6.8, and 37 °C. The values are shown as the mean ± standard deviation, n = 3.

The differences between the maximum $OD_{600}$ in the microbioreactor and the bioreactor could be related to the lack of a control system at the microbioreactor level. The ratio of NaOH consumed and LA produced was approximately 1:5. The *L. paracasei* 6714 fermentation process was controlled for 21 h; however, considering the obtained results, a 14 h control would suffice, based on the LA yield and productivity.

### 3.2. Assessment of Potential for LTA Synthesis

At the end of the fermentation, biomass accounted for approximately 10% of the final working volume (based on experimental observations). Considering *L. paracasei* 6714 batch fermentation, this volume was about 0.387 L per bioreactor. The presence of LTA was investigated, as it is a predominant component of the cell wall in various *Lactobacillus* species. Previous research reported the production of LTA by *L. rhamnosus* [15]; nonetheless, to the best of our knowledge, no previous studies have reported the detection of LTA in *L. paracasei*. Figure 4 shows the PAGE separation and detection of LTA extracted from the biomass. The migration behavior of LTA from both the positive controls from *S. aureus* and *L. paracasei* 6714 was similar. As expected, the LTA band from *L. paracasei* 6714 exhibited a weaker staining with Alcian blue–silver solution. This behavior was normal, as the LTA extract was not purified; a darker band may be observed in the case of a pure sample. So far, this experiment shows a preliminary result that suggests that LTA can be effectively obtained from *L. paracasei* biomass. However, further experiments are necessary to establish LTA production under optimized culture conditions and after the purification of this compound. Such a process should probably target high biomass production, which is proportionally related to LTA production; thus fed-batch fermentation would be recommended.

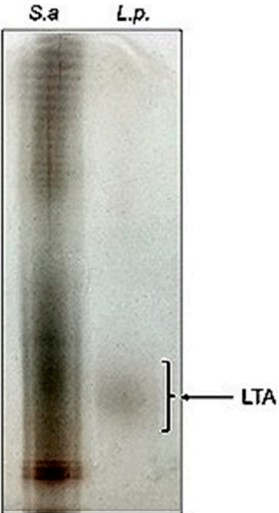

**Figure 4.** Detection of lipoteichoic acid (LTA) from *L. paracasei* 6714-L.p. and *Staphylococcus aureus* (S.a.) (positive control: Sigma Aldrich—PN L2515). LTA was loaded in 20% PAGE gel and visualized using the Alcian blue–silver staining method. The bracket indicates the LTA band of *L. paracasei* 6714.

### 3.3. Assessment of Antagonistic Activity

Another side stream of the DSP that may be underutilized is the one obtained after the nanofiltration process. The nanofiltration step is necessary to remove remaining sugars and proteins that should not enter the mono- and bipolar electrodialysis step due to the high sensitivity of the equipment [36]. The nanofiltration permeate is normally treated as waste; however, it could be used to obtain antimicrobial compounds due to their higher molecular weight. For example, nisin molecules can be recovered using a 3–5 kDa cutoff filter on the retentate flow. In the case of electrodialysis-based DSP, the cutoff is between 150 and 300 Da [31].

The results of the antagonistic activity of the supernatants from the three isolates are shown in Table 2. The surrogate bacteria selected for the study of antagonism represent groups associated with human or animal disease (*Listeria* spp. and *E. coli*) or food spoilage (*P. fluorescens*); these bacteria are commonly found in the environment, and they can be associated with vegetables and food of animal origin. LAB-derived antimicrobial compounds can help to control or decrease the growth of these potentially harmful microorganisms, and they could be recovered from nanofiltration-derived material. The three tested *L. paracasei* isolates caused significant inhibition of all of the surrogate bacterial strains, and the effect was observed even with the most diluted sample (except on *L. paracasei* 6713 and *P. fluorescens*); this could be associated with high antimicrobial activity, even at a low concentration of the bioactive compound. The antimicrobial activity could be attributed to the production of antimicrobial peptides, such as bacteriocins, considering that other metabolites (e.g., LA and AA) were neutralized with NaOH [37]. This is not surprising as several bacteriocins from *L. paracasei* have been identified before. For example, Ye et al. [38] recently reported a new bacteriocin from *L. paracasei* that is active against both Gram-negative and Gram-positive foodborne microorganisms; similar results were observed with the strains from this study. Additionally, Belguesmia et al. [39] reported one *L. paracasei* strain with the capacity to simultaneously synthesize five different bacteriocins, thus confirming the potential of this bacterial group for producing antimicrobial compounds.

**Table 2.** Absorbance values at 600 nm obtained for the determination of the antimicrobial activity of the *L. paracasei* (6712, 6713, and 6714) supernatant against *Escherichia coli*, *Pseudomonas fluorescens*, *Listeria innocua* 4.1, and *Listeria innocua* 5.1.

| Strain | Supernatant Ratio | Absorbance at 600 nm | | | |
|---|---|---|---|---|---|
| | | E. coli 25922 | P. fluorescens 6.2 | L. innocua 4.1 | L. innocua 5.1 |
| | No Supernatant | $1.214 \pm 0.077$ [a] | $0.609 \pm 0.026$ [a] | $0.467 \pm 0.040$ [a] | $0.469 \pm 0.028$ [a] |
| *L. paracasei* (6712) | 1:1 | $0.453 \pm 0.023$ [b] | $0.359 \pm 0.048$ [c] | $0.136 \pm 0.003$ [b] | $0.237 \pm 0.011$ [b] |
| | 1:2 | $0.465 \pm 0.035$ [b] | $0.431 \pm 0.119$ [b,c] | $0.182 \pm 0.007$ [c] | $0.244 \pm 0.010$ [b] |
| | 1:4 | $0.486 \pm 0.013$ [b] | $0.438 \pm 0.049$ [b,c] | $0.196 \pm 0.002$ [d] | $0.257 \pm 0.012$ [b] |
| | 1:8 | $0.694 \pm 0.028$ [b] | $0.569 \pm 0.055$ [a,b] | $0.210 \pm 0.001$ [e] | $0.262 \pm 0.004$ [b] |
| *L. paracasei* (6713) | 1:1 | $0.426 \pm 0.052$ [b] | $0.378 \pm 0.021$ [a] | $0.183 \pm 0.025$ [b] | $0.207 \pm 0.025$ [b] |
| | 1:2 | $0.429 \pm 0.009$ [b] | $0.424 \pm 0.091$ [a] | $0.203 \pm 0.006$ [b,c] | $0.241 \pm 0.024$ [b,c] |
| | 1:4 | $0.469 \pm 0.017$ [b] | $0.537 \pm 0.187$ [a] | $0.216 \pm 0.007$ [b,c] | $0.239 \pm 0.013$ [b,c] |
| | 1:8 | $0.504 \pm 0.017$ [b] | $0.562 \pm 0.176$ [a] | $0.227 \pm 0.012$ [c] | $0.270 \pm 0.006$ [c] |
| *L. paracasei* (6714) | 1:1 | $0.407 \pm 0.045$ [b] | $0.318 \pm 0.037$ [b] | $0.085 \pm 0.004$ [b] | $0.207 \pm 0.013$ [b] |
| | 1:2 | $0.443 \pm 0.055$ [b] | $0.340 \pm 0.046$ [b] | $0.188 \pm 0.004$ [c] | $0.227 \pm 0.010$ [b,c] |
| | 1:4 | $0.487 \pm 0.036$ [b] | $0.363 \pm 0.028$ [b] | $0.190 \pm 0.008$ [c] | $0.239 \pm 0.003$ [c] |
| | 1:8 | $0.694 \pm 0.028$ [b] | $0.394 \pm 0.068$ [b] | $0.226 \pm 0.008$ [d] | $0.358 \pm 0.010$ [b] |

Different letters in column values mean significant differences ($p < 0.05$).

## 4. Conclusions

The identification of LAB isolates with a wide range of metabolic capacities is crucial for the development of new integrated fermentation processes where more products from the DSP steps can be recovered and utilized. The strain *L. paracasei* 6714 was selected for bioreactor fermentation due to its combined higher metabolic capacity; this strain exhibited important antagonistic activity against all of the studied surrogate strains, and, considering its growth rate, $OD_{600}$, presence of LTA in the biomass, and LA production, it is a candidate for multi-product fermentation focused on DSP valorization to make it more profitable. New studies may incorporate optimization experiments to enhance the production of metabolites of interest from selected isolates under ideal conditions (pH, agitation rate, media composition, and temperature) and to establish the cost-effectiveness of the process. The findings of this research reveal the potential of LAB isolates from pineapple silage of agro-industrial residues for the implementation of a multi-product integrated process. This is an important preliminary step, given that Costa Rica is a tropical environment where the diversity and presence of these isolates may be important. To our knowledge, this is the first time that LAB strains from Costa Rican agro-industrial residues have been studied for biotechnology applications, thus opening new opportunities for the sustainable exploitation of these substrates.

**Author Contributions:** Methodology, experimental trial, visualization, data analysis, writing—original draft, review, and editing, J.M.-Z.; experimental trial, visualization, data analysis, writing—original draft, M.D.R.-V.; methodology, conceptualization, visualization, data analysis, writing—original draft, review, and editing, N.B.; writing—original draft, review, and editing J.P.L.-G.; methodology, conceptualization, visualization, data analysis, writing—original draft, review, and editing, J.A.M.-V.; methodology, conceptualization, visualization, data analysis, writing—original draft, review, and editing M.R.-S. All authors have read and agreed to the published version of the manuscript.

**Funding:** This work was supported by the National Center of High Technology of Costa Rica (CeNAT-CONARE Scholarship Program 2019), the Tropical Disease Investigation Center (CIET-UCR), the National Center of Food Science and Technology of University of Costa Rica (CITA-UCR), and the University of Costa Rica (Project B9-457).

**Institutional Review Board Statement:** Not applicable.

**Informed Consent Statement:** Not applicable.

**Data Availability Statement:** Data available upon request.

**Acknowledgments:** Thanks to Melissa Chaves Phillips and Jorge Araya Mattey for their technical support during the LTA analysis.

**Conflicts of Interest:** The authors declare no conflict of interest.

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
