# Peer review of "Potential of New Bacterial Strains for a Multiproduct Bioprocess Application: A Case Study Using Isolates of Lactic Acid Bacteria from Pineapple Silage of Costa Rican Agro-Industrial Residues"

_fermentation, doi:10.3390/fermentation8080361_

Round 1

Reviewer 1 Report

The MS by Montero-Zamora et al. is generally well written and backed up with a lot of characterizations. A few comments for authors to consider:

1. The intorduction could be improved to highlight the research question. At present, I struggle to understand what indeed the goal of this work is. Was is it the growth comparison between strains? What were the multiple product process developed for LA? Where is the novelty in that?

The goal  and novelty of this work is not clear and should be rwritten to highlight those.

2. Figure 4 needs enhancement. As it stands, it is lnot visible or legible at all.

Author Response

Please the attachment

Reviewer 2 Report

This paper describes evaluation of lactic acid bacteria, previously isolated from pineapple processing waste, for their potential to produce lactic acid and acetic acid while also secreting antimicrobial compounds and producing cell wall-bound lipotechoic acid as additional products to increase the value of the process.

Comments     

1. The conclusion states that agro-industrial waste as culture medium should be considered for a biotechnological process and new isolates should be identified and used. Both statements are true; however, this manuscript used culture medium instead of biomass and previously-isolated strains instead of new isolates. Use of a lignocellulosic growth substrate would offer some novelty to the findings.

2. In Section 3.1, please explain how the concentrations and productivity compare to other reports in the literature.

3. Production of antimicrobials should be discussed in context of previous work; what has been observed for other lactic acid bacteria? What are the active compounds likely to be?

4. What was the logic for looking at both homolactic and heterolactic fermenters? Is the goal to produce lactic acid or a mixture of lactic acid and acetic acid?

5. In Figure 4, how much material (cell wall extract/LTA) was loaded in the two lanes? On Line 253, why was a weaker stain expected for L. paracasei 6714? How much LTA production is required in order to make extraction practical or cost-effective; is the comparatively lower amount apparently produced by L. paracasei 6714 worthwhile?

6. Which enantiomer of lactic acid is produced by Lb. paracasei 6714; D, L or a mixture?

 Additional comments:

1.       Line 50: The substrates mentioned are very common sources of lactic acid bacteria. It would be better to say these are non-traditional biorefinery feedstocks, if that is what is meant by this sentence.

2.       What is the acid mentioned on Line 74?

3.       Please provide more information about the type of microplate and microplate reader used to measure growth kinetics.

4.       Lacticaseibacillus paracasei is a genus and species name; however, two of the strains in Figure 2 have different genus and species names. Please correct the Figure 2 legend to clarify.

5.       Line 160 specify the type of assay (antimicrobial susceptibility assays)

6.       Define LA on Line 40 and AA on Line 102. Define LAB on Line 38 and LTA on Line 61. In the abstract, lipotechoic acid does not need to be abbreviated because it is used only once in the abstract.

7.       Line 144: biomass would be reported in grams; 3000 ml indicates media volume

8.       Line 146: PBS usually refers to phosphate-buffered saline, rather than phosphate buffer solution. Please state the composition of PBS used in this study.

9.       Lines 89-90: species parafarraginis and ghanensis should be italicized; genus and species names should be italicized throughout sections 3.1 and 3.2 and in the figure legends

10.   The section numbers skip from 3 Results and Discussion to 5 Conclusions; change Conclusions to 4.

Round 2

Reviewer 2 Report

The authors have handled the comments on the original manuscript; please address these additional comments and questions:

1. Line 241: Is homofermentative correct? That sentence is confusing; I thought homofermenters produce two lactates and heterofermenters produce one lactate and acetate typically.

2. Lines 55-57: Why would poor LA solubility allow recovery of antimicrobial compounds? Please explain or reword the sentence.

3. Line 131: why were the cultures maintained at pH 6.8; I believe LAB usually prefer a lower pH?

4. Line 167: what was the acrylamide:bisacrylamide ratio?

5. Line 282 is there a reference for the 10% value?

6. Section 2.5.2 and Table 2 title and body of table: Strain number for E. coli is missing in the table. Are there strain numbers for P. fluorescens?

7. In Table 2 or methods: For the no supernatant samples, was sterile media added as a control?

8. In the methods section, please be consistent with use of rpm or centrifugal force (g). Preferably g.

9. In the abstract, this phrase on Line 25 is awkward:…evaluate the production of LA, acetic acid and glucose consumption. Change to …evaluate production of LA and acetic acid and consumption of glucose.

10. In the abstract, “the supernatant produced significant inhibition against all bacterial strains.” That sentence is too vague. State the microorganism names or generally describe them i.e. food-associated pathogens, or pathogens associated with human or animal disease.

11. In the abstract, this conclusion is obvious: Agro-industrial residuals may be a good source of LAB isolates with biotechnological potential for multiproduct processes. Consider changing to: LAB isolated from pineapple silage have biotechnological potential for multiproduct processes.

12. Line 71: change since to such as

13. Line 83: Delete “Precisely, an important number of” and instead start the sentence with: “LAB strains with biotechnological potential are…”

14. Line 261: spell out nine

Author Response

Reviewers' comments:

Reviewer #2:

  1. Line 241: Is homofermentative correct? That sentence is confusing; I thought homofermenters produce two lactates and heterofermenters produce one lactate and acetate typically.

Response: The writing was modified to avoid confusion. The new text is as follows: “All the strains from the study were able to produce AA, meaning that they followed the heterofermentative route although with variable LA/AA ratio; L. paracasei strains favored the production of LA in all cases.”

  1. Lines 55-57: Why would poor LA solubility allow recovery of antimicrobial compounds? Please explain or reword the sentence.

Response: low solubility of LA in organic solvents has been reported before and this property could be used to facilitate separation with bacteriocins. This ides is now better explained in the text: “Given that LA has poor solubility in organic solvents, a convenient separation strategy (based on organic extraction) could be applied to easily separate bacteriocins from this compound [7].”

  1. Line 131: why were the cultures maintained at pH 6.8; I believe LAB usually prefer a lower pH?

Response: a pH value of 6.8 was based on previous experimental work performed in our lab. Reference 22 establishes technical detail on the use of this fermentation condition although with a different LAB species. We agree that this may not be the best fermentation condition for L. paracasei and that is why we recommend additional studies to include optimization experiments. A small modification was included in the “Conclusions” section to better express this idea: “New studies may incorporate optimization experiments to enhance the production of metabolites of interest from selected isolates under ideal conditions (pH, agitation rate, media composition, temperature)”

  1. Line 167: what was the acrylamide:bisacrylamide ratio?

Response: The ratio was 30% acrylamide and 6% bisacrylamide. This information was now included in section 2.4.2.

  1. Line 282 is there a reference for the 10% value?

Response: There is no reference as this data was based on our own experimental observations. It is now specified in section 3.2

  1. Section 2.5.2 and Table 2 title and body of table: Strain number for E. coli is missing in the table. Are there strain numbers for P. fluorescens?

Response: The corrections were made accordingly

  1. In Table 2 or methods: For the no supernatant samples, was sterile media added as a control?

Response: Yes, this aspect is now clarified in section 2.5.2.

  1. In the methods section, please be consistent with use of rpm or centrifugal force (g). Preferably g.

Response: The corrections were made accordingly

  1. In the abstract, this phrase on Line 25 is awkward:…evaluate the production of LA, acetic acid and glucose consumption. Change to …evaluate production of LA and acetic acid and consumption of glucose.

Response: The correction was made accordingly

  1. In the abstract, “the supernatant produced significant inhibition against all bacterial strains.” That sentence is too vague. State the microorganism names or generally describe them i.e. food-associated pathogens, or pathogens associated with human or animal disease.

Response: The correction was made accordingly

  1. In the abstract, this conclusion is obvious: Agro-industrial residuals may be a good source of LAB isolates with biotechnological potential for multiproduct processes. Consider changing to: LAB isolated from pineapple silage have biotechnological potential for multiproduct processes.

Response: The corrections was made accordingly

  1. Line 71: change since to such as

Response: The corrections was made accordingly

  1. Line 83: Delete “Precisely, an important number of” and instead start the sentence with: “LAB strains with biotechnological potential are…”

Response: The correction was made accordingly

  1. Line 261: spell out nine

Response: The correction was made accordingly